Mitochondria-targeted triphenylphosphonium-based compounds do not affect estrogen receptor α

Zinovkina Ludmila A. 1 2
Galivondzhyan Alina K. 1
Prikhodko Anastasia S. 2 3
Galkin Ivan I. 3
Zinovkin Roman A. roman.zinovkin@gmail.com 2 3 4
1 Faculty of Bioengineering and Bioinformatics, Lomonosov Moscow State University , Moscow , Russia
2 Institute of Mitoengineering, Moscow State University , Moscow , Russia
3 Belozersky Institute of Physico-Chemical Biology, Lomonosov Moscow State University , Moscow , Russia
4 Institute of Molecular Medicine, Sechenov First Moscow State Medical University , Moscow , Russia
Silva Pedro
Electronic publication date: 2020 Mar 25
Publication date: 2020
Volume: 8
Electronic Location ID: e8803
Received 2019 Nov 8; Accepted 2020 Feb 25
Copyright: ©2020 Zinovkina et al.
Copyright year: 2020
Copyright holder: Zinovkina et al.
License: This is an open access article distributed under the terms of the Creative Commons Attribution License, which permits unrestricted use, distribution, reproduction and adaptation in any medium and for any purpose provided that it is properly attributed. For attribution, the original author(s), title, publication source (PeerJ) and either DOI or URL of the article must be cited.
License URL: https://creativecommons.org/licenses/by/4.0/

Keywords: Triphenylphosphonium, Targeting mitochondria, Estrogen

Funding: The authors received no funding for this work.

==============================
Background

Targeting negatively charged mitochondria is often achieved using triphenylphosphonium (TPP) cations. These cationic vehicles may possess biological activity, and a docking study indicates that TPP-moieties may act as modulators of signaling through the estrogen receptor α (ERα). Moreover, in vivo and in vitro experiments revealed the estrogen-like effects of TPP-based compounds. Here, we tested the hypothesis that TPP-based compounds regulate the activity of ERα.

Methods

We used ERa-positive and ERα-negative human breast adenocarcinoma cell lines (MCF-7 and MDA-MB-231, respectively). Cell proliferation was measured using a resazurin cell growth assay and a real-time cell analyzer assay. Cell cycle progression was analyzed using flow cytometry. Real-time PCR was used to assess mRNA expression of endogenous estrogen-responsive genes. Luciferase activity was measured to evaluate transcription driven by estrogen-responsive promoters in cells transfected with an estrogen response element (ERE)3-luciferase expression vector.

Results

The TPP-based molecules SkQ1 and C12TPP, as well as the rhodamine-based SkQR1, did not increase the proliferation or alter the cell cycle progression of MCF-7 cells. In contrast, 17β estradiol increased the proliferation of MCF-7 cells and the proportion of cells in the S/G2/M-phases of the cell cycle. TPP-based compounds did not affect the induction of transcription of an ERE-luciferase expression vector in vitro, and SkQ1 did not alter the levels of expression of estrogen-dependent genes encoding GREB1, TFF1, COX6, and IGFBP4.

Conclusion

TPP-based compounds do not possess properties typical of ERα agonists.

Introduction

Mitochondria are important target for drug development (Zielonka et al., 2017). Mitochondria-targeted molecules include drug candidates, intracellular probes and sensors (Zinovkin & Zamyatnin, 2019). The drug of interest is usually linked via a carbon linker to the triphenylphosphonium (TPP) cation. The latter effectively confers mitochondria-targeting activity. Lipophilic TPP-based molecules are readily transported across the membranes of cells and mitochondria, and cationic TPP ensures targeting to negatively charged mitochondria through electrostatic attraction. The widely used mitochondria-targeted antioxidants MitoQ and SkQ1 exploit the TPP cation as a vehicle. Despite a wealth of experimental data obtained using TPP-based compounds, numerous molecular features of intracellular signaling induced by TPP-based moieties are unknown.

The TPP cation may possess biological activity. For example, the results of a docking study suggest that TPP-moieties may act as modulators of estrogen receptorα (ERα) (Salisbury & WilliamsJr, 2009). Estrogens are implicated in numerous physiological functions of females and males. The major female sex hormone 17β estradiol (E2) contributes to human development and reproduction. ERs include ERα, ERβ, and G protein-coupled estrogen receptor alpha. Multiple isoforms of ERα and ERβ function as transcription factors. ERs dimerize upon binding to E2, and the complex translocates to the nucleus where it regulates the expression of estrogen-dependent genes. Furthermore, indirect mechanisms operate through multiple co-regulatory proteins (Yasar et al., 2017).

Indirect evidence supports the possibility that the TPP-based antioxidants MitoQ and SkQ1 act through estrogen-like action as follows:

• The mitochondria of MCF-7 and endothelial cells contain functional high-affinity ERs (Pedram et al., 2006). Activation of these receptors by E2 inhibits UV radiation-induced cytochrome C release, decreased mitochondrial membrane potential, and apoptosis via the formation of mitochondrial reactive oxygen species (mROS) (Pedram et al., 2006). MitoQ exhibits the same activities. Moreover, MitoQ and SkQ1 inhibit mROS generation to prevent subsequent apoptosis (Feniouk & Skulachev, 2017), suggesting they directly interact with mitochondrial ERs.

• Estrogen-like activity of SkQ1 is detected in female outbred SHR mice, and long-term oral administration of SkQ1 inhibits the age-related decline of the estrous function (Anisimov et al., 2008).

• SkQ1 consumption by female rats prolongs proestrus duration, typical of E2 (Chistyakov et al., 2012).

• ERα regulates multiple NF-κB pathway components to control inflammatory responses (Kovats, 2015), and SkQ1 regulates NF-κB activity in endothelial cells (Zinovkin et al., 2014).

• E2 (Brotfain et al., 2016), MitoQ (Zhou et al., 2018), and SkQ1 (Silachev et al., 2015) exert neuroprotective effects in a model of traumatic brain injury. 

Experimental and clinical evidence link sustained exposure to estrogens with increased risk of developing breast cancer (Russo & Russo, 2006). Certain mitochondria-targeted antioxidants are the subject of clinical trials (Zinovkin & Zamyatnin, 2019). Eye drops containing SkQ1 are used to cure dry eye syndrome (Visomitin) and MitoQ is freely available as an antioxidant supplement.

TPP-based mitochondria-targeted compounds may possess estrogen-like activities, and it is critically important to investigate this possibility. Here we tested the hypothesis that TPP-based compounds such as SkQ1 regulate the activity of ER α. For this purpose, we tested the TPP-based compounds C12TPP and SkQ1 (plastoquinone conjugated to C12TPP). The mitochondria-targeted antioxidant SkQR1, which lacks the TPP moiety, served as a negative control. The structures of these cations are shown in Fig. 1.

Figure 1 Mitochondria-penetrating cations used in the present study.

(A) C12TPP. (B) SkQ1. (C) SkQR1.

Materials & Methods

Reagents

E2 and its inhibitor were 17 β-Estradiol (E2758; Sigma) and fulvestrant (I4409; Sigma), respectively. The protonophore was 2,4-dinitrophenol (DNP; D198501; Sigma). The cell culture reagents included phenol-red-free DMEM media (Paneco, Russia); charcoal-stripped fetal bovine serum (FBS; F6765; Sigma); FBS (SV30160.03; HyClone) and DMEM medium (Gibco). AlamarBlue reagent (DAL1025) was purchased from Biosource, and propidium iodide (PI) was purchased from MP Biomedicals, France. SkQ1, SkQR1, and C12TPP were synthesized as bromide salts at the Belozersky Institute of Physico-Chemical Biology.

Cell culture

The human breast adenocarcinoma cell lines (ERα-positive MCF-7 and ER-, progesterone receptor-, and erb-b2 receptor tyrosine kinase 2-negative [triple-negative] MDA-MB-231) were obtained from the Russian Collection of Cell Cultures (Institute of Cytology, St Petersburg, Russia). Cells were maintained in DMEM with 10% FBS at 37 °C in an atmosphere containing 5% CO2. The media was replaced at least 6 days before an experiment, with phenol-red-free DMEM with 5% charcoal stripped FBS to exclude activation of ERs by phenol red and serum components. The images of MCF-7 cells treated with test compounds were acquired by phase-contrast microscopy using Axioplan (Zeiss) microscope.

Resazurin cell growth assay

Cells (approximately 5,000 cells per well) were added to the wells of 96-well plates. After 2 days, the medium was replaced with fresh medium with or without the test compound. After 48 h, cell viability was determined using a resazurin assay with AlamarBlue reagent (2.75 mM). Fluorescence was measured after 3.5 h at 544/590 nm. Each experiment was repeated five or six times.

Real-time cell analyzer (RTCA) assay

The xCELLigence system (Roche Applied Sciences, Basel, Switzerland) was used to evaluate cell growth rate and viability. The system calculates a Cell Index (CI) value defined as (Rn-Rb)/Rb, where Rn is the impedance in the well at time n, and Rb is the background impedance. The cells (20,000 cells/well) were cultured in E-plates for 24 h, and the medium was replaced with fresh medium supplemented with 10 nM E2, 20 nM SkQ1, 20 nM C12TPP or 20 nM SkQR1. For the first 2 h, CI values were determined every 5 min, and thereafter at 15 min intervals.

Cell cycle analysis

PI was used to analyze the cell cycle distribution (Galkin et al., 2014). Briefly, cells (200,000 cells per well) were added to the wells of 6-well plates, incubated for 24 h and then for 48 h with the test compounds or controls, harvested using free-EDTA trypsin, washed with PBS, and fixed in cold 70% ethanol overnight at 4 °C. The cells were subsequently stained with PI containing RNAse A and the samples were analyzed using a flow cytometer (Beckman Coulter FC500).

Transfection and luciferase reporter assay

The estrogen-responsive element (ERE)3–Luc reporter construct (Hall & McDonnell, 1999) containing three copies of the vitellogenin ERE was purchased from Addgene (plasmid no. 11354). The plasmid was purified using a Plasmid midi kit (QIAGEN, USA). Cells were grown to 80%–90% confluence and transiently transfected with TransIT-LT1 Transfection Reagent (Mirus Bio, USA). Firefly luciferase activity was measured using a Luciferase Assay System (Promega, USA), and luminescence was measured using a LKB 1251 Luminometer. Luminescence was normalized to the total protein content measured using the Bradford assay.

RNA purification and real-time PCR

Total RNA was purified from MCF-7 cells using a Quick-RNA MiniPrep Kit according to the manufacturer’s protocol (Zymo Research, USA). Purified RNA was reverse transcribed using SuperScript III reverse transcriptase (Thermo Fisher, USA) and used for real-time PCR performed using an iCycler iQ amplifier (Bio-Rad, USA) with EVA Green Master Mix (Syntol, Russia) according to the manufacturer’s instructions. Primer sequences are listed in Table S1. The efficiencies of the primers were within 88%–100%. The relative expression levels of the target genes was determined using the ΔΔCt method with the calculated primer efficiencies. RPL32 mRNA encoding ribosomal protein L32 served as a reference.

Statistical analysis

Data were analyzed using the one-way ANOVA with post-hoc Tukey HSD, or with an unpaired Student’s t- test. P < 0.05 was considered to indicate a significant difference.

Results

Effect of TPP-based compounds on the growth of an estrogen-dependent human breast cancer cell line

We measured cell proliferation in the presence of 0.2 to 20 nM C12TPP and SkQ1. SkQR1 served as a negative control (Fig. 1). Higher concentrations (e.g., ≥200 nM) significantly inhibited cell growth (Fig. S1). E2 (10 nM) served as a positive control, and fulvestrant (10 nM) served as an ERα antagonist. ER-positive MCF-7 cells were incubated for 48 h with the test compounds with or without fulvestrant, and cell growth was measured using the resazurin test.

Treatment of MCF-7 cells with 10 nM E2 significantly (p < 0.02 [ANOVA]) enhanced cell proliferation (Fig. 2). The TPP-based compounds and SkQR1 did not significantly affect cell proliferation. However, 20 nM SkQ1, 0.2 nM C12TPP, 20 nM C12TPP and 20 nM SkQR1 increased resazurin fluorescence by approximately 10% compared with untreated cells (Fig. 2). This increase was not significant and concentration-independent. Fulvestrant significantly (p <0.04 [ANOVA]) lowered resazurin fluorescence in cells treated with 10 nM E2, and the fluorescence of most samples decreased, although the differences were not significant.

Figure 2 Effect of mitochondria-targeted compounds on the proliferation of ER-positive MCF-7 cells.

Cells (approximately 5,000 cells per well) were incubated for 48 h with (A) SkQ1; (B) C12TPP; (C) SkQR1, and DNP. Data are expressed as the percentage of the control cells +/ − standard deviation (SD), n = 5–6. P values were determined using the one-way ANOVA with post-hoc Tukey HSD. *p, E2 compared with the control; #p, E2 and E2 + fulvestrant.

The same experiments employing ERα-negative MDA-MB-231 cells did not reveal a significant effect of E2 or the test compounds (Fig. 3). However, resazurin fluorescence increased after treatment with 2 nM SkQ1, 20 nM SkQ1, 20 nM C12TPP, or 20 nM SkQR1 but not with E2 (Fig. 3). These data suggest that resazurin fluorescence may not correlate with the number of cells but with a slight change in mitochondrial metabolic activities potentially induced by TPP cations. In agreement with this supposition, resazurin fluorescence was slightly enhanced by the mitochondrial protonophore 2,4-dinitrophenol (12 M DNP) when added to MCF-7 or MDA-MB-231 cells (Figs. 2C, 3C). We concluded therefore that the resazurin assay is inapplicable to the estimation of cell proliferation in the presence of mitochondria-targeted compounds. Thus, other assays are required to unambiguously measure the effect of the TPP-based compound on cell proliferation.

Figure 3 Effect of mitochondria-targeted compounds on the proliferation of ER-negative MDA-MB-231 cells.

The cells were cultured and treated as shown in Fig. 2 and proliferation was analyzed using a resazurin assay. (A) SkQ1; (B) C12TPP; (C) SkQR1, and DNP. Data are expressed as a percentage of the control cell cultures +/ − SD, n = 4.

The potential of TPP-based compounds to affect the proliferation of ER-positive MCF-7 cells was tested using an RTCA assay (Fig. 4). The test compounds, except E2, did not significantly affect the CI value. E2 started to increase the CI value of MCF-7 cells approximately 36 h after its addition.

Figure 4 Proliferation of ER-positive MCF-7 cells treated with mitochondria-targeted compounds.

CI values were dynamically monitored using a xCELLigence system after the administration (arrow) of 10 nM E2 (Estrogen), 20 nM SkQ1 (SkQ1), 20 nM C12TPP (C12TPP), or 20 nM SkQR1 (SkQR1). The average values of triplicate experiments are plotted against treatment time.

Effect of TPP-based compounds on the cell cycle distribution of estrogen-dependent breast cancer cells

Cell proliferation is characterized by increases in the numbers of cells in the S, G2, and M phases. To determine the distribution of cells among these phases, we performed flow cytometric analysis of MCF-7 cells treated for 48 h with the test compounds and then with PI. E2 treatment increased the numbers of cells in the S, G2, and M phases (Fig. 5) There were no detectable changes in cell cycle distribution after treatment of the MCF-7 cells with 20 nM SkQ1, 20 nM C12TPP, and 20 nM SkQR1 (Fig. 5).

Figure 5 Cell cycle analysis of ER-positive MCF-7 cells.

Cells were treated as shown in Fig. 2. The cell cycle was analyzed using flow cytometry of PI-stained cells. (A–E) Percentages of cells in S/G2/M phases. (A) “Control,” untreated cells. (B) “Estrogen,” 10 nM E2. (C) “SkQ1,” 20 nM SkQ1. (D) “C12TPP,” 20 nM C12TPP. (E) “SkQR1,” 20 nM SkQR1. (F) The percentages of cells distributed in SubG1, G1/G0, S, and G2/M phases, and the percentage of polyploid cells.

Effect of TPP-based compounds on the transcription driven by an estrogen-responsive promoter

Direct binding of an agonist to ERα activates ERE-dependent transcription. To assess the effect of TPP-based cations on ERE-dependent transcription, we transfected MCF-7 cells with the luciferase reporter construct ERE3–Luc, followed by incubation for 24 h with the test compounds. Only E2 increased ERE-dependent transcription and corresponding luciferase-induced luminescence (approximately 5-fold, p = 0.04 [Student’s t-test]; Fig. 6). The E2 antagonist fulvestrant partially inhibited this effect (p = 0.21 [Student’s t-test]). Treatment of the transfected cells with 20 nM SkQ1, 20 nM C12TPP, and 20 nM SkQR1 revealed no significant change in luminescence (Fig. 6).

Figure 6 TPP-based compounds do not enhance expression driven by estrogen-responsive promoters in MCF-7 cells.

Cells were transfected with an ERE3–Luc reporter construct. Test compounds were added 24 h after transfection followed by measurement of luminescence 24 h later. Luminescence was normalized to the total protein content. Data are expressed as the percentage of untreated transfected cells +/− SD, n = 3; *p = 0.04, untreated control vs E2-treated cells; #p = 0.04, untreated vs cells treated with both E2 and fulvestrant (unpaired Student’s t-test).

To further evaluate the possible effect of SkQ1 on the transcriptional activity of the estrogen-responsive promoters, we measured the expression in MCF-7 cells of E2 target genes encoding the following proteins: trefoil factor 1 (TFF1); growth regulating estrogen receptor binding 1 (GREB1); cytochrome c oxidase subunit VI (COX6); insulin-like growth factor binding protein 4 (IGFBP4). These genes were chosen using available mRNA expression data for MCF-7 cells treated with E2 (Yamaga et al., 2013) and relative mRNA expression was estimated using real-time PCR.

E2 treatment of MCF-7 cells increased the levels of the transcripts of all target genes (Fig. 7). The simultaneous addition of the estrogen antagonist fulvestrant with E2 partially prevented the increase in the expression of TFF1 and GREB1. Interestingly, the same effect of fulvestrant on the expressions of TFF1 and GREB1 was observed in vehicle and SkQ1-treated samples, indicating the presence of a small amount of estrogen mimetics in the culture medium.

Figure 7 Analysis of gene expression in MCF-7 cells.

MCF-7 cells were treated as shown in Fig. 2, and the levels of mRNAs encoding GREB1, IGFBP4, COX6, and TFF1 were normalized to the reference RPL32 mRNA. The relative expression of the target genes in the untreated control cells is expressed as 100%. The values represent the mean +/− SD, n = 4. P-values were calculated using the unpaired Student’s t-test; #p, fulvestrant (Fulv) vs corresponding sample without fulvestrant; *p, samples without fulvestrant vs control.

Treatment with any concentration of SkQ1 did not significantly influence the mRNA levels of all E2 target genes. Nevertheless, in some experiments 0.2 nM and 20 nM SkQ1 increased IGFBP4 expression (p = 0.047 and p = 0.13, respectively [Student’s t-test]). However, IGFBP4 expression was not decreased in E2-treated cells treated with fulvestrant, and IGFBP4 mRNA levels therefore cannot be considered to reliably indicate signaling downstream ERα.

Discussion

Testing the effects of any compound on the proliferation of ERα-positive cells is commonly employed to evaluate estrogenic properties. There are several methods available to measure cell proliferation rates. The most popular methods, the MTT assay and the resazurin test, rely on the measurement of activity of mitochondrial enzymes. The resazurin assay measures the activity of mitochondrial dehydrogenases, which convert the nonfluorescent dye resazurin to the strongly-fluorescent dye resorufin. However, the mitochondrial function greatly affects the NAD(P)H-based viability assays, including the resazurin test (Aleshin et al., 2015). Mitochondrial uncoupling, in particular, leads to the overestimation of the MTT test results (Maioli et al., 2009). Moreover, numerous penetrating cations such as SkQ1, C12TPP, and SkQR1 induce mitochondrial uncoupling mediated by free fatty acids (Severin et al., 2010; Antonenko et al., 2011). In an agreement with these data, we observed a moderate statistically insignificant increase in the resazurin fluorescence in MCF-7 cells treated with the mitochondrial penetrating cations or with the “classical” protonophore DNP (Figs. 2C, 3C).

We found it interesting that the test compounds induced slightly higher resazurin fluorescence in MCF-7 than in triple-negative MDA-MB-231 cells (Figs. 2 and 3). We speculate that the uncoupling of oxidative phosphorylation in mitochondria and corresponding activity of mitochondrial dehydrogenases may differ between these cell lines because the basal oxygen consumption rate and proton leakage significantly differs between these cells (Radde et al., 2016). Generally, the MTT and resazurin tests are suitable for the compounds that do not influence mitochondrial function. Therefore, these tests cannot be applied to the study of TPP-based cations with mitochondrial uncoupling activities.

We therefore used the RTCA assay to measure the effects of TPP-based compounds on the proliferation of MCF-7 cells. We found that these compounds did not significantly influence the CI value, which is directly proportional to the quantity, size, and attachment forces of the cell (Fig. 4). Further, microscopic examination did not reveal morphological changes induced by E2 or the test compounds (Fig. S2). Moreover, the CI value is proportional to the cell number measured using the RTCA assay, and TPP cations therefore did not significantly influence the proliferation of MCF-7 cells. Furthermore, we did not observe a detectable effect of TPP-based compounds on cell cycle distribution (Fig. 5), showing that TPP cations did not enhance the proliferation of ER α-positive MCF-7 cells.

Despite the absence of detectable stimulatory effects of TPP on the proliferation of MCF-7 cells, we cannot exclude the direct binding of TPP to ERα induces downstream signaling that regulates ERE-dependent gene expression. Specifically, we found no detectable effect of TPP cations on the luciferase activity of MCF-7 cells transfected with the ERE3-luciferase reporter (Fig. 6). However, the structures of the promoter regions of ERE-dependent genes is much more complex than the three ERE of the reporter. Further, E2 may indirectly modulate gene expression via the interaction of ER with transcription factors such as NF-κB, AP-1, and Sp-1, which, in turn, bind their cognate DNA regulatory elements (Marino, Galluzzo & Ascenzi, 2006). We therefore tested SkQ1 as a representative TPP-based compound for its ability to regulate the expression of the ER-dependent genes GREB1, TFF1, COX6, and IGFBP4. We found that SkQ1, at any concentration tested, did not significantly upregulate the expression of these genes (Fig. 7). Nevertheless, the expression of IGFBP4 mRNA varied, and the effect of SkQ1 on its expression requires further investigation. Our present data therefore lead us to conclude that SkQ1 did not directly affect ERα and its associated downstream signaling activity related to the expression of ER-dependent genes.

In agreement with this conclusion, several lines of evidence, described below, indicate that TPP-based compounds do not possess proestrogenic effects:

• SkQ1 enhances human neutrophils apoptosis (Andreev-Andrievskiy et al., 2016; Vorobjeva et al., 2017), and E2 has the opposite effect (Molloy et al., 2003).

• Though global gene expression data for TPP-based compounds are scarce, available data on oral administration of MitoQ to wild-type mice indicate that MitoQ treatment does not affect a specific process or signaling pathway mediated by sex hormones (Rodriguez-Cuenca et al., 2010).

• In mouse and rat strains, estrogen increases the incidence of tumorigenesis in several organs (Vollmer, 2003). In three mouse strains studied, prolonged SkQ1 treatment does not affect spontaneous carcinogenesis (Yurova et al., 2011). Moreover, SkQ1 does not increase the development of spontaneous tumors in BALB/c mice (Manskikh et al., 2014).

Theoretically, mitochondria-targeted compounds may possess estrogen-like activities that are not due to TPP cation. For example, certain plant quinones are phytoestrogens (Davis, 2002), and plastoquinone and ubiquinone are essential components of SkQ1 and MitoQ antioxidants, respectively. There are no reports, to our knowledge, about the ER-related activities of these quinones or the rhodamine 19 moiety of SkQR1. Our data further confirm the absence of estrogen-like activities of plastoquinone and the quinone of rhodamine 19.

The present study has the following limitations: first, we did not investigate the direct binding of the compounds to ERα. However, our data strongly suggest low or no binding of the test compounds to ERα. Second, we only used cell lines derived from breast cancers. Thus, the corresponding primary cells may react in a different way. Third, we cannot exclude the possibility that in vivo environmental conditions may greatly differ from the in vitro experimental conditions used here. Hypothetical coregulatory factors in tissues and organs may be involved in indirect estrogen-dependent signaling pathways together with TPP cations. Thus, indirect involvement of TPP-based compounds in estrogen signaling cannot be fully excluded.

Conclusions

For the first time, to our knowledge, we show here that TPP-based cations do not possess properties of typical ERα agonists. Our finding therefore opens new possibilities for safe use of TPP-based compounds in preclinical and clinical applications.

Supplemental Information

Table S1 The sequences of the PCR primers are provided from 5′to 3′ends

Click here for additional data file.

File S1 Resazurin fluorescence in cell proliferation experiments for MCF-7

Raw data (with background subtracted) with resazurin fluorescence obtained in the experiments with MCF-7 cells. Each data point indicates individual fluorescence values (column “Fluorescence€”) for the compounds studied (column “Compound”).

Click here for additional data file.

File S2 Resazurin fluorescence in cell proliferation experiments for MDA-MB-231

Raw data (with background subtracted) with resazurin fluorescence obtained in the experiments withMDA-MB-231 cells. Each data point indicates individual fluorescence values (column “Fluorescence€”) for the compounds studied(column “Compound”).

Click here for additional data file.

File S3 The raw data of luminescence reads of MCF-7 cells transfected with ERE–Luc reporter construct

The column “Compound” indicates the compounds studied; the column “Protein conc. mkg/mL” indicates the concentration of the proteins measured in the Bradford assay; column “Luminescence” indicates raw data obtained by measuring luminescence on a luminometer LKB 1251 using Luciferase Assay System.

Click here for additional data file.

File S4 The raw data for real-time PCR as the threshold data (Ct) for the gene expression study

The column “Sample” indicates the compounds studied; the columns with gene names indicates corresponding genes; the column Ct€ indicates raw threshold data (Ct).

Click here for additional data file.

File S5 Resazurin fluorescence in cell proliferation experiments for MCF-7 cells

Raw data (with background subtracted) with resazurin fluorescence obtained in the pilot experiment with MCF-7 cells. Each data point indicates individual fluorescence values (column “Fluorescence €”) for the compounds studied (column “Compound”).

Click here for additional data file.

Figure S1 Pilot experiment testing the effect of TPP-based compounds and SkQR1 on the proliferation of MCF-7 cells

MCF-7 cells (approximately 5,000 cells per well) were incubated for 48 h with the test compounds at the indicated (nM) concentrations. (A) SkQ1, (B) C12 TPP, and (C) SkQR1. Cell viability was measured according to resazurin fluorescence. Data are expressed as the percentage of the control untreated cells +/ − SD, n = 5 technical replicates. The p-values were determined using unpaired Wilcoxon tests with Bonferroni correction for multiple measurements compared with the untreated control (Vehicle).

Click here for additional data file.

Figure S2 Microscopic observations of MCF-7 cells treated with test compounds

MCF-7 cells were treated as indicated in Fig. 4, and images were acquired using phase-contrast microscopy after 72 h. (A) Control cells. (B) 10 nM E2. (C) 20 nM SkQ1. (D) 20 nM C_12TPP. (E) 20 nM SkQR1. Bar, 50 µm.

Click here for additional data file.

Additional Information and Declarations

Competing Interests

Author Contributions

Data Availability

The authors declare there are no competing interests.

Ludmila A. Zinovkina and Ivan I. Galkin performed the experiments, authored or reviewed drafts of the paper, and approved the final draft.

Alina K. Galivondzhyan performed the experiments, prepared figures and/or tables, and approved the final draft.

Anastasia S. Prikhodko analyzed the data, authored or reviewed drafts of the paper, and approved the final draft.

Roman A. Zinovkin conceived and designed the experiments, analyzed the data, prepared figures and/or tables, and approved the final draft.

The following information was supplied regarding data availability:

The raw measurements of resazurin fluorescence in cell proliferation experiments for MCF-7, the raw measurements of resazurin fluorescence in cell proliferation experiments for the MDA-MB-231 cell line, the raw data of luminescence reads, the raw data for real-time PCR as the threshold data (Ct), the results of the pilot experiment on MCF-7 cell proliferation with raw measurements of resazurin fluorescence, the sequences of the PCR primers, the histogram plotted with the results of the pilot experiment testing the effect of TPP-based compounds and SkQR1 on the proliferation of MCF-7 cells and the microscopic observations of MCF-7 cells treated with test compounds is available in the Supplemental Files.

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
