# Peer review of "Mitochondria-targeted triphenylphosphonium-based compounds do not affect estrogen receptor α"

_PeerJ, doi:10.7717/peerj.8803_

## Round 0.1 · original submission · Minor Revisions

Our reviewers generally approved of your work but there are several issues that require discussion. I look forward to receiving a new version of your work!

Reviewer 1 ·

Basic reporting

The article is well written.

Experimental design

I suggested to check if the compounds bind to ERα.
I suggested to add PPT as selective estrogen receptor ERα agonist in addition to E2.

Validity of the findings

It is hard to get the conclusion.

·

Basic reporting

1. Figures should be relevant to the content of the article, of sufficient resolution, and appropriately described and labeled.
2. Several original data (Lines 171, 249 and 262) should be given as supporting information even though they are not critical for this research.

Experimental design

The exposure concentrations used in this research should be more reasonable.

Validity of the findings

The conclusions should be appropriately stated, should be connected to the original question investigated, and should be limited to those supported by the results.

Additional comments

Zinovkina and coworkers report TPP-based SkQ1 and C12TPP, as well as the rhodamine-based SkQR1 do not directly interact with ERα and do not induce estrogen-like changes in MCF-7 cells. The present study confirms the general fact that TPP-based compounds are safe to use. I like the idea and methodological concept of the study. However, some major concerns that in my opinion markedly reduce the impact of the study. These can be found below.
Major concern:
(1) In the present study, the MCF-7 cells (estrogen receptor alpha positive) and MDA-MB-231 cells (triple-negative, also known as estrogen receptor alpha negative) were selected as in vitro models to investigate the influence of triphenylphosphonium (TPP)-based compounds and the underling mechanisms as well. By assessing the cell proliferation, cell cycle progression and gene transcription, the authors showed that mitochondria-targeted TPP-based compounds have no direct action on the estrogen receptor α. Herein, no significant differences were found between MCF-7 and MDA-MB-231 cells based on the resazurin assay, which should be deeply explained. As you known, other than the estrogen receptor alpha, both cells are expressed other membrane estrogen receptors (mERS) such as estrogen receptor alpha 36 (ERα36) and G protein coupled receptor (GPR30). Those mERs may also participate in TPP-based compounds-mediated proliferative effects, which should be further investigated.
(2) In the present study, all experiments were conducted 5-6 replicates independently, so the statistical significance should be evaluated and be marked accurately in the figures.
(3) The authors showed TPP-based compounds at higher concentrations significantly inhibited cell proliferation, but these original data and underlying mechanisms were not presented.
(4) The authors showed that TPP-based compounds had no significant effects on the cell cycle distribution unlike that of E2. What I'm focusing on is that dose these TPP-based compounds also have no effects on the gene expression of several cell cycle-regulated protein. In fact, different cell cycle stages (SubG0, G0/G1, S and G2/M phases) should be investigated separately.

Minor issues:
(1) Different concentrations of C12TPP were used in MCF-7 and MDA-MB-231 cells (Line 175-185), is it correct? If correct, the authors should give some basis.
(2) How many duplicate samples were used in the real-time cell analyzer assay? From Figure 4, I cannot find this formation.
(3) English should be improved on some places - for instance, the sentences in Reagents are somehow weird and I recommend rewriting it for better clarity. This also applies to other portions of the text, such as Lines 203-204 and Lines 215-217.

·

Basic reporting

This is a descriptive study where the authors demonstrated that triphenylphosphonium-based compounds do not act as estrogen receptor α modulators. Manuscript is in general well delineated, very objective and has appropriate references. Results are in agreement with the methodology and the hypothesis proposed. Although results are negative, it is important to the field to have this information.

Experimental design

Authors added as control a transform cell line; however, results might be different in primary naive cells.

Validity of the findings

No comment

Additional comments

No comments

---

## Round 0.2 · accepted · Accept

I am satisfied with your responses and I accept your manuscript.